# Evaluation of Global Land Use–Land Cover Data Products in Guangxi, China

Xuan Hao [1,2,3], Yubao Qiu [2,3,4,*], Guoqiang Jia [2,3,4], Massimo Menenti [5,6], Jiangming Ma [1] and Zhengxin Jiang [3]

1   Guangxi Key Laboratory of Landscape Resources Conservation and Sustainable Utilization in Lijiang River Basin, Guangxi Normal University, Guilin 541006, China
2   International Research Center for Big Data for Sustainable Development Goals, Beijing 100094, China
3   Key Laboratory of Digital Earth, Aerospace Information Research Institute, Chinese Academy of Sciences, Beijing 100094, China
4   China-ASEAN Regional Innovation Center for Big Earth Data, Nanning 530022, China
5   Faculty of Civil Engineering and Earth Sciences, Delft University of Technology, 2628 Delft, The Netherlands
6   State Key Laboratory of Remote Sensing Science, Aerospace Information Research Institute, Chinese Academy of Sciences, Beijing 100101, China
*   Correspondence: qiuyb@aircas.ac.cn

**Abstract:** Land use–land cover (LULC) is an important feature for ecological environment research, land resource management and evaluation. Although global high-resolution LULC data sets are booming, their regional performances were still evaluated in limited regions. To demonstrate the local applicability of global LULC data products, six emerging LULC data products were evaluated and compared in Guangxi, China. The six products used are European Space Agency GlobCover (ESAGC), ESRI Land Use–Land Cover (ESRI–LULC), Finer Resolution Observation and Monitoring of Global Land Cover (FROM–GLC), the China Land Cover Dataset (CLCD), the Global Land Cover product with Fine Classification System at 30 m (GLC_FCS30) and GlobeLand30 (GLC30). Reference data were obtained from the local government statistical yearbook and high-resolution remote sensing images on Google Earth. The results showed that CLCD, ESRI–LULC and GLC30 were found to agree well with the forest reference data, with the highest correlation coefficient of 0.999. For the cropland areas, GLC30, CLCD and ESAGC agreed well with the reference data, and the highest correlation coefficient was 0.957. Combined with the comparison with the high-resolution images obtained by Google Earth, we finally concluded that ESAGC, CLCD and GLC30 can best represent the LULCs in Guangxi. Furthermore, the spatial consistency analysis showed that three or more products identified the same LULC type as high as 96.98% of the area. We suggest that majority voting might be applied to global LULC products to provide fused products with better performances on a regional or local scale to avoid the error caused by a single data product.

**Keywords:** land use–land cover; data inter-comparison; spatial consistency analysis; fusion; forest and cropland

## 1. Introduction

Land use–land cover (LULC) information is an important basis for land resource evaluation and land development. Specific land types also have profound effects on hydrological, ecological and carbon cycle processes [1,2]. As a medium of interaction between human systems and other systems, LULC types on land surface are direct manifestations of the interaction between human activities and the natural environment [3,4]. The state of the economy also depends on land use patterns and the intensity of land use to some extent [5]. With the acceleration of industrialization and urbanization since the late 20th century, the spatial pattern of land use in China has undergone significant changes. Land use systems are under threat due to climate change and human activities. Assessment of LULCs at the global, regional and local levels is essential in order to better monitor food security,

provide basic information for sustainable development policy research, and serve climate and environmental change research [6–12].

Traditional methods of acquiring land cover information mainly rely on on-site surveys, which are time-consuming and costly but give results that have a high degree of accuracy. Since the 1990s, the imaging capability of the various remote sensing systems that provide data with different temporal and spatial resolutions has greatly improved. Satellite remote sensing has many functions, including rapid, large-scale and periodic acquisition of surface information. It provides the basis for the acquisition, integration and in-depth analysis of global and regional land cover information [13]. A variety of global remote sensing-based land cover products are currently available, including University of Maryland Land Cover Classification 1998 created by the UMD [14]; IGBP (International Geosphere Biosphere Program) DISCover produced by the US Geological Survey [15]; GLC2000 data produced by the European Union's Joint Research Center [16]; the MODIS (Moderate-resolution Imaging Spectroradiometer) land cover data product produced by Boston University [17]; and the GLOBCOVER product, which is produced by the European Space Agency [18]. These data sets play a very important role in many fields, including physical geography, environment and global change studies [19]. Using satellite image data, a number of researchers has conducted qualitative and quantitative surveys of the potential of the entire country by categorizing it into several geographic ecotourism zones based on its geographical environment and land use–land cover (LULC) characteristics [20]. Based on the vegetation formation group and vegetation formation scales use the European Space Agency's annual land cover data and geo-information TUPU analysis to investigate vegetation succession direction, succession speed and succession sequence in Inner Mongolia from 1992 to 2018 [21]. To examine the interaction between environmental factors, LULC and conflict in South Kordofan, Sudan, some scholars analyzed the changes of LC in the study area in the past 30 years (1984–2014) [22]. However, poor resolution at both temporal and spatial scales limits their further application at regional scales. Some researchers have conducted cross-comparisons between Google's Dynamic World, ESA's (European Space Agency) World Cover and ESRI's (Environmental Systems Research Institute) Land Cover data sets and also evaluated the accuracy of these products. The results have shown that these three global LULC maps give nearly equal estimates for the areas covered by the water, built-up area, tree and crop LULC classes for 2020. They recommend that the use of global LULC products should involve critical evaluation of their suitability with respect to the application purpose [23]. In recent years, high-resolution remote sensing and big data have been well-developed. Along with the emergence of remote sensing land cover products with high spatial resolution, these include ESA GlobCover (ESAGC) [24], Finer Resolution Observation and Monitoring of Global Land Cover (FROM–GLC) [25] and GlobeLand30 (GLC30) [26]. However, current classification–discrimination models tend to be based on random point data samples from around the world. Although the spatial resolution of these land cover products has been well-improved, the performance of global data at regional scales remains to be evaluated. When the same study was conducted by different research teams, the vast majority of the differences between the results could not be attributed to a specific factor. The conclusion of seemingly objective quantitative procedure needs some evaluation in practical application [27].

At regional scales, the quantitative evaluation of land cover information is of relevance to social, economic and environmentally sustainable development, and changes in land cover are closely related to the changes in the processes such as the Earth's carbon and water cycles [28]. Although increasing attention is being paid by policymakers and the scientific community to the regional application of large-scale LULC products [29–32], related research is still limited in some regions. For example, scholars made a qualitative and quantitative analysis and comparison of the five existing kinds of land cover data for China and the surrounding areas based on a spatial consistency analysis and by sampling the high resolution images in Google Earth. The results revealed large areas where there was disagreement between the five land cover data sets, and the overall consistency was

low [33]). Researchers conducted a unit-by-unit comparison test between an Italian land cover map based on a field survey and GlobeLand30 data and assessed the quality of the GlobeLand30 data classification by obtaining a confusion matrix and related statistical indicators. The authors concluded that the overall accuracy of GlobeLand30 was about 80% within the scope of the study area [34]. Experts compared and evaluated four land cover data sets covering China by using China's 1:100,000 large-scale land cover map as reference data. Problems including local labeling error, low labeling accuracy and consistency and a high mixing error were found. The results also showed that all four data sets were deficient to some extent and did not meet the standard needed for surface modeling [35]. Some researchers adopted a sample design method that integrated field inspection and grid sampling to evaluate a 2015 1:100,000 land cover data set of China and concluded that the data set had a high accuracy in Henan [36]. Some academics evaluated the classification accuracy of four global land cover data sets (version2 of the IGBP global land cover data set, the 2001 MODIS land cover map, the global land cover map produced by the University of Maryland and the land cover map produced by the Global Vegetation Monitoring unit of the European Commission Joint Research Centre (GLC2000)) for China based on classification consistency and a spatial consistency analysis. The authors concluded that none of the four data sets were sufficiently accurate to meet the needs of simulations of land surface processes and suggested that a land cover fusion method based on the existing multiple sources of land cover classification information should be developed [37]. Professors used the high-precision CHINA-2010 land cover data set to verify the spatial consistency of GlobeLand30. It was found that, at the scale of individual Chinese provinces, GlobalLand30 was highly accurate, but that its spatial consistency with CHINA-2010 decreased as the complexity of the land cover increased. It was also found that the accuracy of GlobalLand30 was low in areas of elevation transition [24].

In summary, LULC is the foundation for ecological environment research, land resource management and evaluation, and has profound effects on hydrological, ecological and carbon cycle processes as well as human well-being and economy. Although global high-resolution LULC data sets are booming, their regional performances were still evaluated in limited regions, which hinders the wide application of global LULC data sets. In this study, the performance of six widely used global and national LULC data sets in Guangxi was evaluated. The cropland and forest area data were compared with local yearbook statistics, and the data sets were validated by comparison with high-resolution remote sensing images obtained from Google Earth. Finally, we established a fused land cover data set for Guangxi to obtain more spatially accurate land cover data in Guangxi. It provides a case for LULC evaluation and application in the southwestern subtropical region.

## 2. Data and Methods

### 2.1. Study Area

Guangxi Zhuang Autonomous Region (Guangxi) lies in southern China (at latitudes 20.9–26.4°N and longitudes 104.47–112.07°E) within a low-latitude subtropical region where water vapor and heat are abundant all year round. The annual precipitation ranges from 723.9~2983.8 mm, and the annual mean temperature is between 17.6~23.8 °C (http://www.gxzf.gov.cn/mlgxi/gxrw/zrdl/t1003584.shtml, accessed on 10 February 2023). Moreover, unique karst landforms can be found in the region. The terrain is mountainous with little flat land; overall, the central parts of the region form a basin with the areas around it having higher elevations. Thus, the surface processes are complex and dynamic. Guangxi also serves as a bridge in the logistics hub between China and the Association of Southeast Asian Nations (ASEAN). In recent years, Guangxi has experienced increasing utilization of its land resources and increasing economic development. Detecting patterns and changes in regional LULC is of great importance to efforts supporting environmental protection and sustainable social development. Figure 1 shows the geographical location and elevation map of the study area.

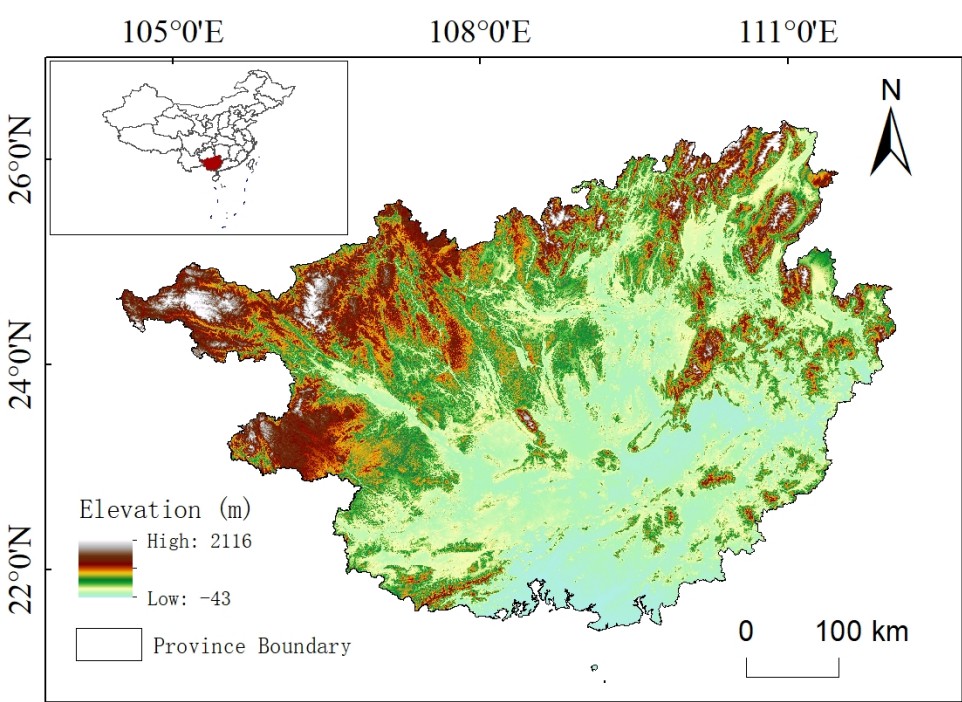

**Figure 1.** Geographical location and elevation map of Guangxi, China.

### *2.2. LULC Products and Reference Data*

#### 2.2.1. LULC Data Products

In this study, six widely used remote sensing-based land cover data sets were selected for assessment of their performance in Guangxi. The data sets included ESA GlobCover (ESAGC), ESRI Land Use/Land Cover (ESRI–LULC), Finer Resolution Observation and Monitoring of Global Land Cover (FROM–GLC), the China Land Cover Dataset (CLCD), the Global Land Cover product with Fine Classification System at 30 m (GLC_FCS30) and GlobeLand30 (GLC30).

ESA GlobCover is a global land cover data product that has a spatial resolution of 10 m (URL: https://esa-worldcover.org/en, accessed on 12 March 2022). It is produced by the European Space Agency and based on Sentinel-2 optical remote sensing imagery. The land cover data are produced using the decision-tree classification method, and the overall accuracy of the product is 75% [38].

ESRI Land Use–Land Cover is a global 10-m land cover data set that was developed by ESRI based on Sentinel-2 surface reflection data in six bands. (URL: https://www.arcgis.com/apps/instant/media/index.html?appid=fc92d38533d440078f17678ebc20e8e2, accessed on 12 March 2022). The product is constructed using a deep learning model, and the overall accuracy is 85% [39].

FROM–GLC10 is a product based on Sentinel-2 images that were acquired in 2017. (URL: http://data.ess.tsinghua.edu.cn/fromglc2017v1.html, accessed on 12 March 2022). From these images, this global 10-m land cover data set was produced using a random forest classifier. The product was validated using multi-seasonal samples with uniform global coverage that were obtained from Landsat 8 images acquired in 2014 and 2015; further interpretation was carried out by experts. The overall accuracy of the data set is 72.35% [25].

The China Land Cover Dataset is a 30-m land cover classification data set of China that was obtained by constructing several temporal metrics using 335,709 Landsat images on GEE and inputting them into the random forest classifier (URL: https://doi.org/10.5281/zenodo.4417810, accessed on 12 March 2022). This data set has good spatiotemporal consistency. Based on visual interpretation of 5,463 samples, the overall accuracy is calculated as 79.31% [40].

GLC_FCS30 is a fine-resolution (30-m) product that can be used for the dynamic monitoring of global surface land cover over the period from 1985 to 2020 (URL: http://data.casearth.cn/sdo/detail/5fbc7904819aec1ea2dd7061, accessed on 12 March 2022). This data set was produced using all available Landsat data from the period 1984 to 2020 with an update every five years. The overall accuracy of this product is 82.5% [41].

GlobeLand30 is a 30-m global surface land cover data set developed by the China National Basic Geographic Information Center based on Landsat, HJ-1 and GF-1 satellite data (URL: http://mulu.tianditu.gov.cn/mapDataAction.do?method=globalLandCover, accessed on 12 March 2022). The overall accuracy of GlobeLand30 V2020 data is 85.72% [26].

In this study, data from 2020 were used for evaluation of these data sets, except in the case of FROM–GLC, for which the latest available data dates from 2017. Further information about each data set is shown in Table 1. All information of the data accuracy is derived from research reports of data producers or data references of original authors. The accuracy was evaluated and calculated by the data producers with a large number of fieldwork validation samples. More detailed information is available in their references.

**Table 1.** Details of the six land cover data sets used in this study.

| Name of Data Set | ESAGC | ESRI–LULC | FROM–GLC | CLCD | GLC_FCS30 | GLC30 |
|---|---|---|---|---|---|---|
| Major Research and Development Unit | European Space Agency | Environmental System Research Institute, ESRI | Tsinghua University | Wuhan University | Aerospace Information Research Institute, Chinese Academy of Sciences | National Geomatics Center of China |
| Calculation Platform | Google Earth Engine | Microsoft Planetary Computer | Google Earth Engine | Google Earth Engine | Google Earth Engine | — |
| Data Sources | Sentinel-2 | Sentinel-2 | Landsat-8 | Landsat-5 | Landsat-8 | Landsat, HJ-1, GF-1 |
| Extraction Method | Decision tree Classifier | Deep learning classification | Supervised classifierRandom forest classification | Supervised classifier | Supervised classifier | Unsupervised classification-POK classification method |
| Verification Method | Independent verification | Confusion matrix | Sample evaluation | Visual interpretation of samples | Confusion matrix | Landscape shape index sampling model samples |
| Spatial Coverage | Global | Global | Global | China | Global | Global |
| Spatial Resolution | 10 m | 10 m | 10 m | 30 m | 30 m | 30 m |
| Date | 2020 | 2020 | 2017 | 2020 | 2020 | 2020 |
| Overall Accuracy | 75% | 85% | 72.76% | 79.31% | 82.5% | 85.72% |

### 2.2.2. Forest and Cropland Data

In this study, we evaluated the different land cover data sets by comparing the individual product data with the statistics for specific land cover types provided by regional yearbooks. The detailed statistical data were obtained from the *Guangxi Statistical Yearbook*, which includes annual city-level forest area statistics and county-level crop-area statistics [42]. URL: http://tjj.gxzf.gov.cn/, accessed on 5 January 2022. These statistics are mainly based on annual statistical reports, surveys and censuses produced by the statistics bureaus of different administrative levels in Guangxi. To ensure that the remote sensing-based land cover data and statistical data were from the same period, statistical data from

the 2020 yearbook were used for the comparisons. Where 2020 statistical data were missing, data from 2019 were used.

### 2.3. Consolidation of LULC Classification

Different land cover classification systems are used for the six selected land cover data sets. Thus, it was necessary to first produce a unified classification system. We reclassified the six land cover data sets into ten land cover types by following the existing land cover reclassification system [43–45]. The reclassified data were than validated using Google Earth sample points that were manually identified based on the definitions of the different land cover types in each data set. We unified the similar land cover types among different data sets by comparing their definition, such as "tree cover", "trees", "forests", to a unified type as "forest". Moreover, we merged several fine classifications, such as "rainfed-cropland", "irrigated cropland" as "cropland". The final land cover classification system consisted of the following classes: cropland, forest, shrubland, grassland, water, wetland, impervious surface, bare land, ice and snow, and other. Table 2 shows the correspondence between the new classes and the original classes in the different land cover products.

**Table 2.** Table showing correspondence between land cover classes after reclassification and the original classes used for the six land cover products.

| Reclassification Results | Results of the Original Data Land Cover Classification | | | | | |
|---|---|---|---|---|---|---|
| | ESAGC | ESRI–LULC | FROM–GLC | CLCD | GLC_FCS30 | GLC30 |
| Cropland | Cropland | Crops | Cropland | Cropland | Rainfed cropland/Herbaceous cover/Tree or shrub cover (orchard)/Irrigated cropland | Cultivated land |
| Forest | Tree cover | Trees | Forest | Forest | Open evergreen broadleaved forest/Closed evergreen broadleaved forest/Open deciduous broadleaved forest/Closed deciduous broadleaved forest/Open evergreen needle-leaved forest/Closed evergreen needle-leaved forest/Open deciduous needle-leaved forest/Closed deciduous needle-leaved forest/Open mixed leaf forest/Closed mixed leaf forest | Forest |
| Grassland | Grassland | Grass | Grassland | Grassland | Grassland | Grassland |
| Shrubland | Shrubland | Scrub/shrub | Shrubland | Shrub | Shrubland/Evergreen shrubland/Deciduous shrubland | Shrub land |
| Wetland | Herbaceous wetland | Flooded vegetation | Wetland | Wetland | Wetlands | Wetland |
| Water | Permanent water bodies | Water | Water | Water | Water body | Water bodies |
| Impervious surface | Built-up | Built Area | Impervious surface | Impervious | Impervious surfaces | Artificial Surfaces |

**Table 2.** *Cont.*

| Reclassification Results | Results of the Original Data Land Cover Classification | | | | | |
|---|---|---|---|---|---|---|
| | **ESAGC** | **ESRI–LULC** | **FROM–GLC** | **CLCD** | **GLC_FCS30** | **GLC30** |
| Bare land | Bare/sparse vegetation | Bare ground | Bareland | Barren | Lichens and mosses/Sparse vegetation/Sparse shrubland/Sparse herbaceous/Bare areas/Consolidated bare areas/Unconsolidated bare areas | Bare land |
| Snow/Ice | Snow and Ice | Snow/Ice | Snow/Ice | Snow/Ice | Permanent ice and snow | Permanent snow and ice |
| Other | Mangroves/ Moss and lichen | Clouds | Tundra | | | Tundra |

### 2.4. Analysis Methods

#### 2.4.1. Correlation Analysis

To analyze the similarity between the different land cover data sets, we calculated the proportional area covered by the different land cover types for the six land cover data sets. The correlation between the area series of different land cover types in any two data sets was then used as a measure of the similarity between the data sets. Correlation coefficient is widely used in various fields as a common index to measure the degree of linear correlation between two random variables [46,47]. The formula used for the correlation coefficient was:

$$CC = \frac{\sum_{k=1}^{r} (x_k - \overline{\chi})(y_k - \overline{y})}{\sqrt{\sum_{k=1}^{r} (\chi_k - \overline{x})^2 \sum_{k=1}^{r} (y_k - \overline{y})^2}}, \tag{1}$$

where $k$ is the index of the land cover type, $r$ is the total number of land cover types, $x_k$ is the percentage area of land cover type $k$ in data set $x$, $y_k$ is the percentage area of land cover type $k$ in data set $y$, $\overline{x}$ is the mean percentage area of all land cover types in data set $x$ and $\overline{y}$ is the mean percentage area of all land cover types in data set $y$.

#### 2.4.2. Accuracy Assessment

By analyzing the size of the difference between the evaluated data and the reference data, we could determine the degree to which the data being evaluated deviates from the reference data, that is, the accuracy of the evaluated data [47,48]. To verify the accuracy of the different LULC data products, we compared the areas of the forest and cropland in the six land cover data sets with the relevant data from the statistical yearbook. The yearbook data included forest area data for 14 city-level administrative units in Guangxi and cropland area data for 111 county-level administrative units.

The accuracy of the LULC data products was assessed by calculating the error coefficient:

$$\varepsilon_i = \left| \frac{K_i - N_i}{N_i} \right|, \tag{2}$$

where $i$ represents the unit for which the error was calculated—a city-level unit in the case of the forest data and a county-level unit in the case of the cropland data. $K_i$ is the area of cropland or forest for the selected land cover data set in unit $i$, and $N_i$ is the area of cropland or forest in unit $i$ as given in the yearbook. The smaller the error coefficient, the more similar the data in the land cover product and the reference data and the higher the reliability and accuracy of the land cover product.

2.4.3. Spatial Consistency Analysis

In order to accurately illustrate the spatial differentiation of a certain land cover type, we adopted the spatial consistency analysis method to determine whether the land cover type is consistently indicated by different products, and we drew an intuitive thematic map pixel by pixel. We overlaid the six land cover data sets by using ArcGIS (v.10.5) software and examined whether the land cover type in the same grid was the same in the different data sets. The mapping grid used for the ESAGC was considered as the benchmark. We counted the times that each pixel was detected as having the same land cover type in the six data products. The larger the number, the higher the spatial consistency of different data sets. A value of 4–6 was considered to represent a high degree of consistency, a value of 3 to represent medium consistency and a value of less than 3 to represent a low degree of consistency.

## 3. LULC Data Product Reclassification

### 3.1. Spatial Distribution of Land Cover Types in Reclassified LULC Data Sets

Figure 2 shows the spatial distribution of the land cover types in the six data sets after reclassification. It can be seen that the spatial patterns of land cover are similar for all six data sets, with forest, followed by cropland, being the most widely distributed cover type. The forest areas are concentrated around the outer parts of Guangxi at relatively high altitudes, whereas the areas of cropland are scattered across the relatively flat areas in the center of the region. The impervious surface class is mainly found in the main cities and the surrounding areas in the form of small blocks. Despite the similarities, there are still some differences in the spatial distribution of the classes in the six data sets; the differences between GLC_FCS30 and the other data sets are the most obvious.

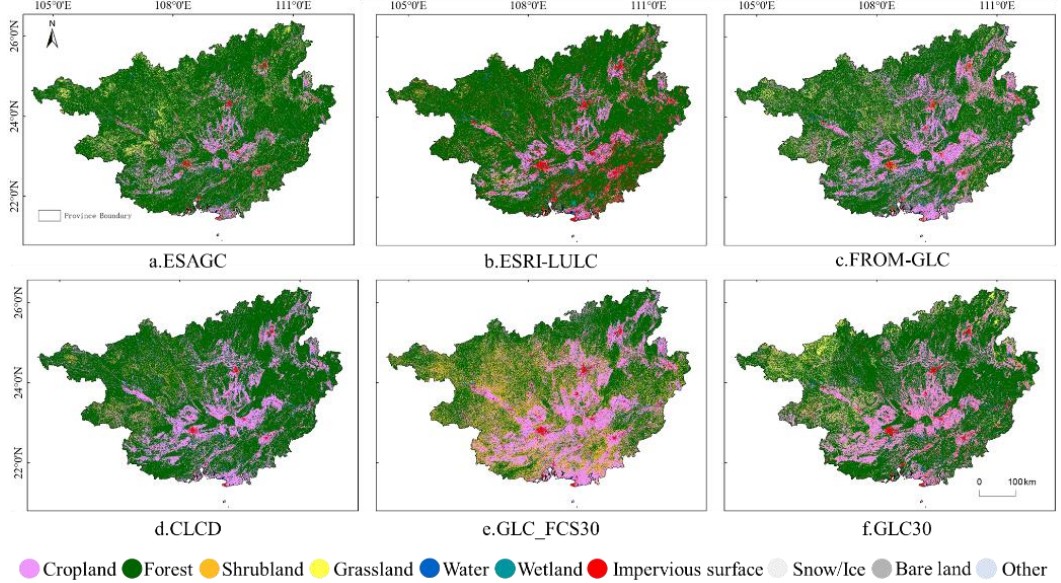

**Figure 2.** Map showing the spatial distribution map of land cover types in Guangxi in the different land cover products after reclassification in 2020.

### 3.2. Area Composition and Similarity Analysis of Land Cover Types

Figure 3 shows the percentage areas covered by the different land cover types in the six data sets. Overall, there is a good level of consistency among the six data sets. The forest cover type covers the largest area in Guangxi—the percentage area ranges from 46.72% to 75.93%, according to the different data sets. This is followed by the cropland class with values ranging from 11.09% to 28.56%. Among the different data sets, ESAGC has the largest forest area and GLC_FCS30 has the smallest. For cropland, the percentage areas in ESAGC and ESRI–LULC are lower than in the other four data sets. The impervious surface

percentage ranges from 1.24% to 8.23%, with the ESRI–LULC percentage being significantly higher than that of the other data sets. For water, the percentage areas in the six data sets are similar and in the range 1.0–2.1%. FROM–GLC and GLC share the highest percentage for grassland, while the percentage of shrubland in GLC_FCS30 is significantly higher than that in the other data sets.

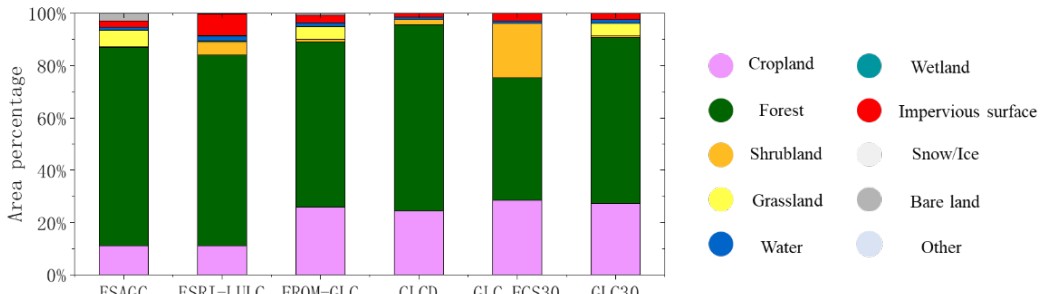

**Figure 3.** The composition of the six land cover products in Guangxi by cover type.

The spatial correlation here represents the degree of similarity between the total area of land cover types in different regions and the official statistical data in each land cover product. It was found that the correlation between the area percentages for the different land cover types across the six land cover data sets was good, with correlation coefficients ranging from 0.828 to 0.999 (Table 3). The correlation between FROM–GLC and GLC30 was the best—giving a correlation coefficient of 0.999; the correlation between ESAGC and GLC_FCS30 was the weakest—giving a correlation coefficient of 0.828. Based on the average correlation coefficient of all pairings, GLC_FCS30 is significantly lower than all others, indicating that this data set differs significantly from the other five data sets. This may partly result from the differences in the classification systems used by GLC_FCS30 and the other products.

**Table 3.** Correlation coefficients between the six different land cover products.

|  | ESAGC | ESRI–LULC | FROM–GLC | CLCD | GLC_FCS30 | GLC30 |
|---|---|---|---|---|---|---|
| ESAGC | - | 0.989 | 0.966 | 0.974 | 0.828 | 0.959 |
| ESRI–LULC | 0.989 | - | 0.961 | 0.974 | 0.858 | 0.954 |
| FROM–GLC | 0.966 | 0.961 | - | 0.996 | 0.907 | 0.999 |
| CLCD | 0.974 | 0.974 | 0.996 | - | 0.909 | 0.995 |
| GLC_FCS30 | 0.828 | 0.858 | 0.907 | 0.909 | - | 0.909 |
| GLC30 | 0.959 | 0.954 | 0.999 | 0.995 | 0.909 | - |
| mean value | 0.943 | 0.947 | 0.966 | 0.970 | 0.882 | 0.963 |

## 4. Evaluation of the Accuracy of LULC Products

### 4.1. Comparison and Evaluation Based on Forest Data in the Statistical Yearbooks

Figure 4 shows the distribution of forest by city in Guangxi according to the six remote sensing-based land cover data sets and the statistical yearbook data. It can be seen that the cities of Baise, Hechi, Guilin, Liuzhou and Nanning have the largest areas of forest. Compared with the statistical yearbook data, ESAGC and CLCD have larger areas for forest; the area of forest in GLC_FCS30 is the smallest. The data for the other three data sets are broadly similar.

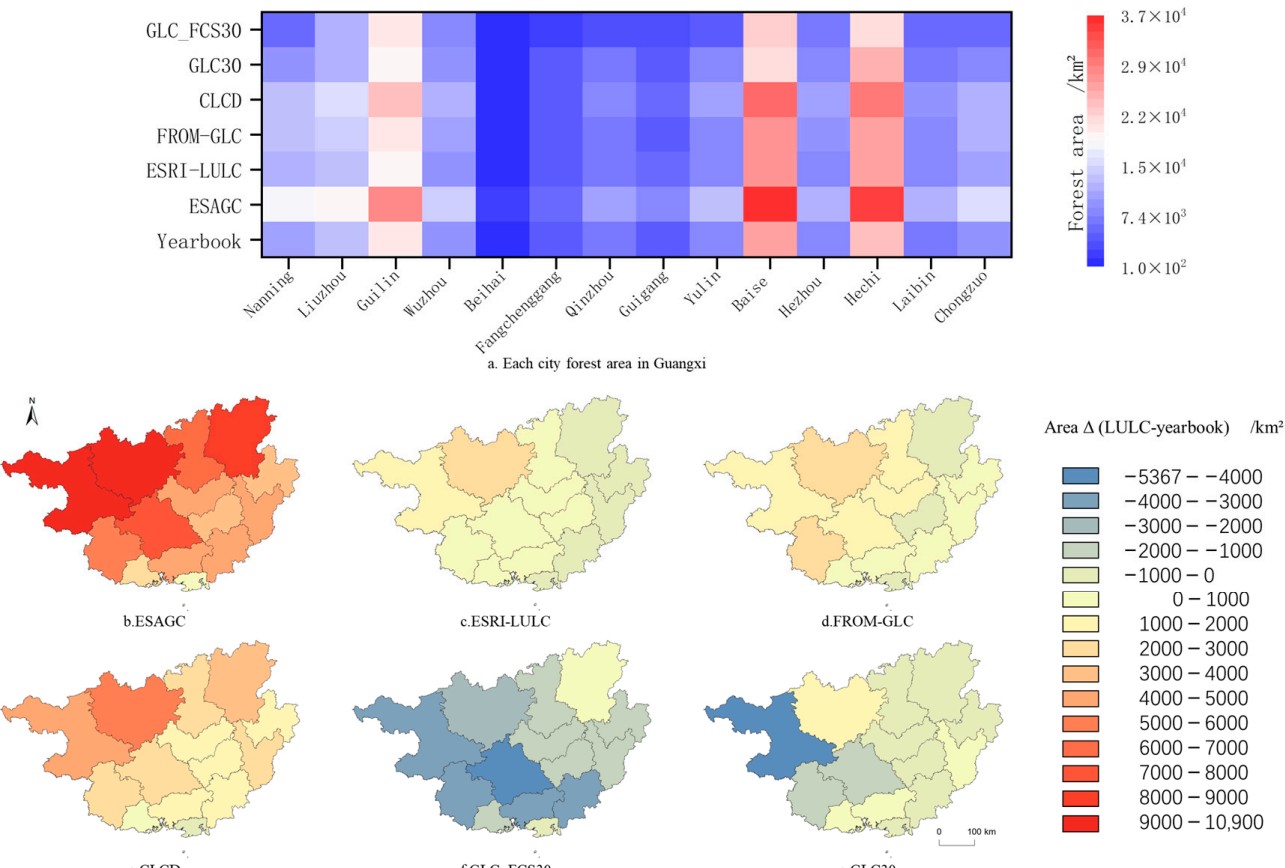

**Figure 4.** Area of forest at the municipal level according to the six land cover products and the statistical yearbook data: (**a**) heat diagram showing the area of forest area in each city in Guangxi for each data set; (**b**–**g**) the difference in forest area between the different land cover data sets and the yearbook values.

Based on the images extracted from the products, the total forest area of each city in Guangxi was compared with the corresponding forest area of each city recorded in the statistical yearbook. A scatter plot for analysis was drawn. Figure 5 consists of scatter plots showing the city-level forest area data from the six land cover data sets plotted against the yearbook data. There is generally good agreement between the land cover data sets and the yearbook data, and the correlation coefficient is greater than 0.98 in each case. ESAGC, ESRI–LULC, FROM–GLC and CLCD generally overestimate the area of forest at the city-level, whereas GLC_FCS30 and GLC30 have lower areas of forest than the statistical yearbooks. The forest areas in the LULC, FROM–GLC, CLCD and GLC30 data sets are close to the values given in the yearbooks.

Figure 6 shows box plots of the error coefficient between the city-level forest data in the different land cover products and the yearbook data. The ESRI–LULC and GLC30 data sets have the lowest error coefficients and thus appear to perform best in terms of forest classification.

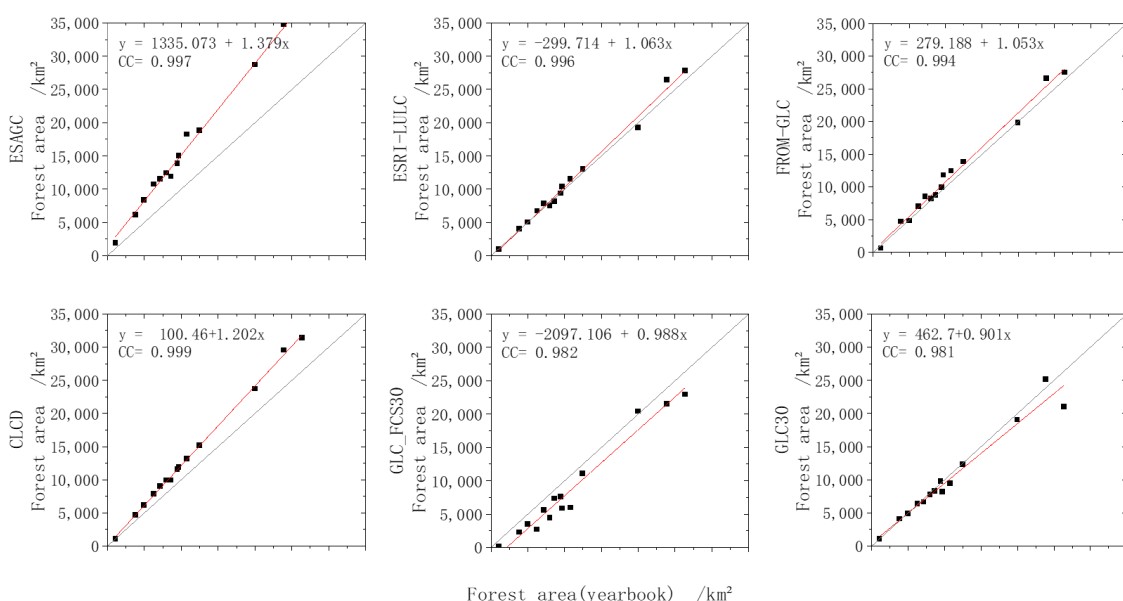

**Figure 5.** Scatter plots showing a comparison of the forest area at the municipal level in Guangxi between the six land cover data sets and the statistical yearbook data.

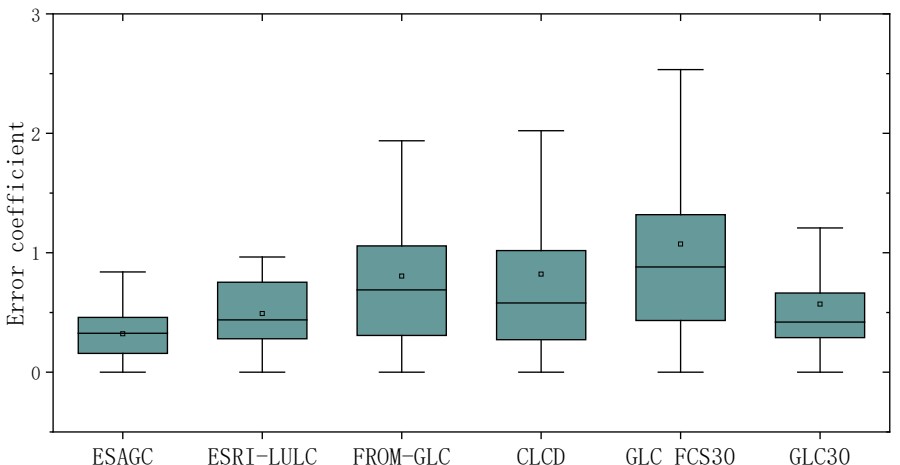

**Figure 6.** Box plots showing the relative error in forest area between the city-level forest data in the different land cover products and the yearbook data for Guangxi.

*4.2. Comparison and Evaluation Based on Cropland Data in the Statistical Yearbooks*

Figure 7 shows the distribution of cropland at the county level in Guangxi, according to the six remote sensing-based land cover data sets and the statistical yearbook data. It can be seen that Baise County, Heng County, Quanzhou County, Teng County, Hepu County, Lingshan County, Guiping City, Bobai County and Xingbin District have the largest areas of cropland. Compared with the statistical yearbook data, ESAGC and ESRI–LULC have smaller areas of cropland; the area of cropland in GLC_FCS30 is the largest. The data for the other three data sets are broadly similar.

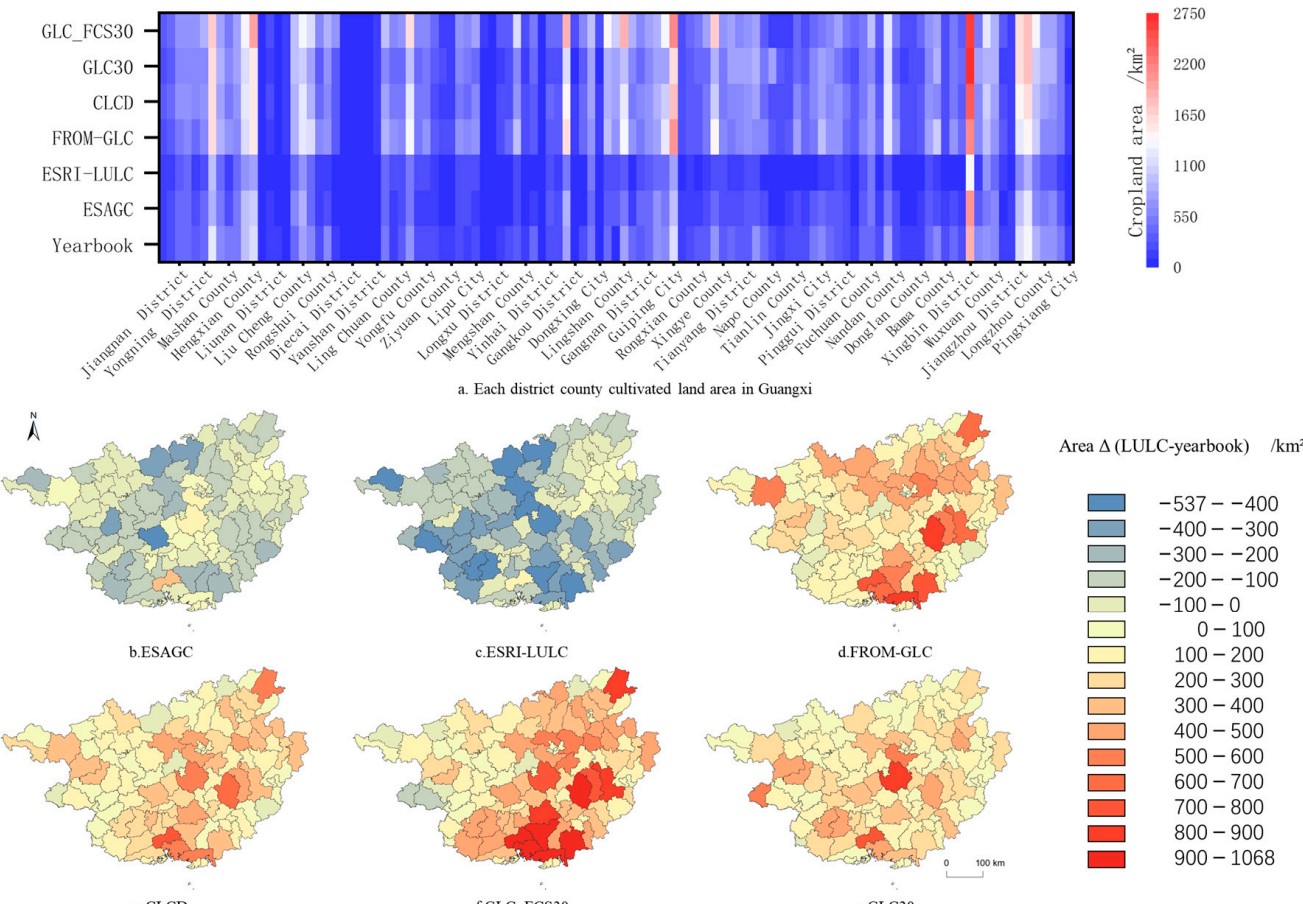

**Figure 7.** Area of cropland at the county-level according to the six land cover products and the statistical yearbook data: (**a**) heat diagram showing the area of cropland in each county in Guangxi for each data set; (**b–g**) the difference in cropland area between the different land cover data sets and the yearbook values.

Figure 8 consists of scatter plots showing the county-level cropland area data from the six land cover data sets plotted against the yearbook data. As was the case with the forest data, there is generally good agreement between the land cover data sets and the yearbook data, with the correlation coefficients being at least 0.88 in each case. ESAGC, FROM–GLC, CLCD, GLC_FCS30 and GLC30 generally overestimate the area of cropland at the county level, whereas the areas of cropland in ESAGC and ESRI–LULC are lower than the areas given in the statistical yearbooks. The cropland areas in the ESAGC, FROM–GLC, CLCD and GLC30 data sets are close to the values given in the yearbooks.

Figure 9 shows box plots of the error coefficient between the county-level cropland area data in the different land cover products and the yearbook data. In this case, the ESAGC, GLC30 and ESRI–LULC data sets have the lowest error coefficients and thus appear to perform best in terms of cropland classification.

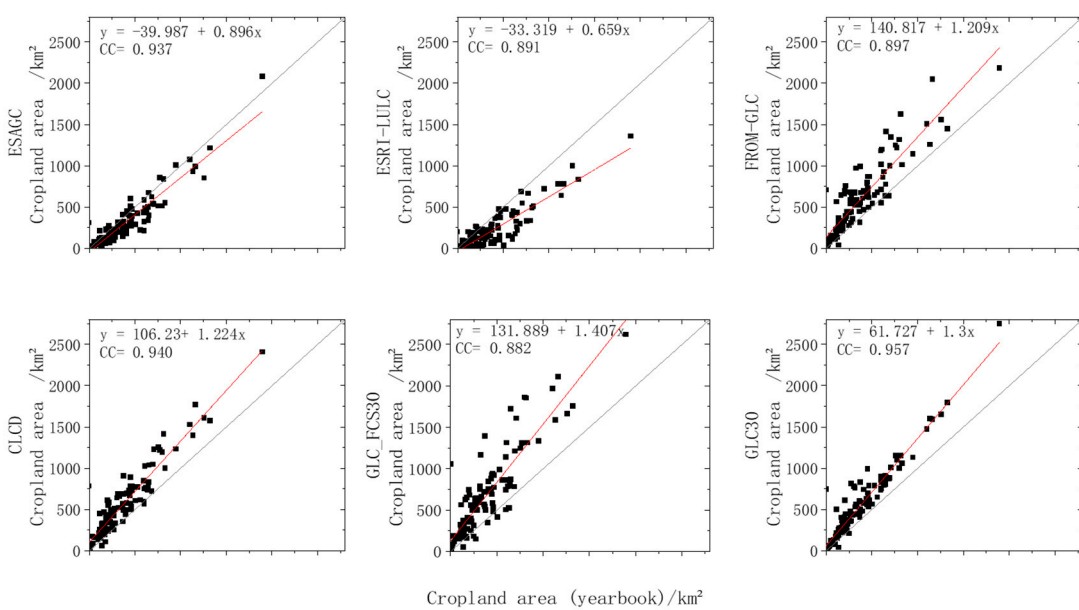

**Figure 8.** Scatter plots showing a comparison of the cropland area at the county level in Guangxi between the six land cover data sets and the statistical yearbook data.

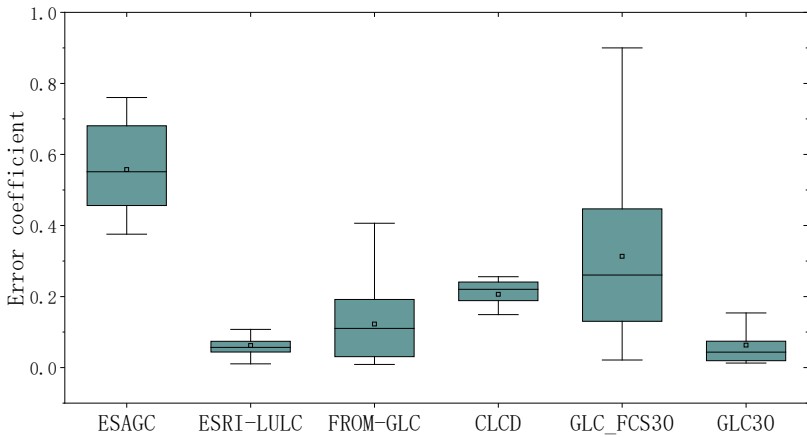

**Figure 9.** Box plots showing the relative error in cropland area between the county-level cropland data in the different land cover products and the yearbook data for Guangxi.

*4.3. Comparison and Verification Based on High-Resolution Remote Sensing Images*

Google Earth uses a compilation of satellite imagery and photography from many different angles, at different resolutions and with different filters to create a 3-D planet Earth. It provides the latest satellite imagery having spatial resolution less than 1 m and is also validated by a large number of fieldwork samples. We did not find any papers or reports from Google Company to show the accuracy of Google Earth data. However, it almost could be seen as the truth with high-resolution and huge fieldwork information. The images on Google Earth have been widely used for land use map preparation and validation (e.g., [49]).

The experimental research in this paper is mainly carried out by visually comparing the main land cover types, i.e., forest and cropland, and is validated with high-resolution remote sensing images. According to Figure 3, the difference between the forest land area data of different products and the statistical yearbook data is concentrated in the mountainous area with large relief in the north of Guangxi. According to Figure 6, the areas with large difference in cultivated land area data are concentrated in the gentle areas with more cultivated land in the central and southern Guangxi. Thus, we selected random validation samples, mainly with forests in the north of Guangxi, samples mainly with

cropland in the central and southern Guangxi. For further validation, we selected random validation samples with complex land cover types. We have three sample groups, i.e., samples mainly with forest, samples mainly with cropland and samples with complex land cover types. For each group, six locations are randomly selected. A total of 18 sample areas in Guangxi were selected for verification (Figure 10a); these included areas of cropland, forest and complex land cover types. The size of these areas was 1 km × 1 km. For these areas, the six remote sensing-based land cover maps were compared with high-resolution remote sensing images obtained from Google Earth (Figure 10b–d).

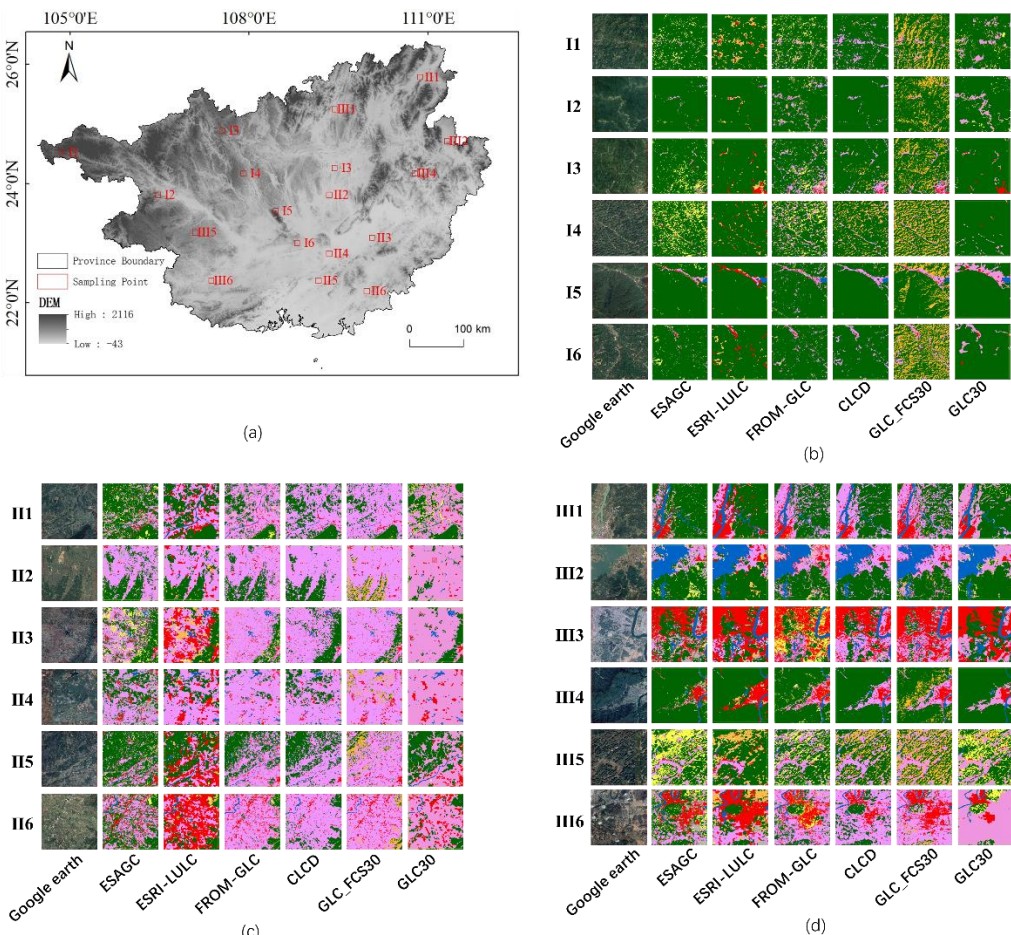

**Figure 10.** Partial comparison of the six land cover products with Google Earth imagery: (**a**) locations of sample areas used for the comparison together with a digital elevation map for Guangxi, (**b**) details of the comparison for sample areas with large differences in forest area at the municipal level, (**c**) details of the comparison for sample areas with large differences in cropland area at the county level and (**d**) details of the comparison for sample areas with complex land cover types.

It can clearly be seen that, of the data sets with a spatial resolution of 10 m, the ESAGC data set contains smaller areas consisting of different land cover types than the ESRI–LULC data set. The FROM_GLC data set contains obvious rectangular boundaries as a result after the downscaling process. Among the other three data sets with a 30-m spatial resolution, in the CLCD, river discontinuity occurs in the identification of smaller rivers. The GLC30 data is unable to detect small areas covered by impervious surfaces. After a further consideration of these results together with the results of the comparison with the forest and cropland data given in the statistical yearbooks, we concluded that the overall performance of ESAGC, CLCD and GLC30 is superior to that of the other data sets. The experimental research in this paper is mainly carried out by comparing the main land cover types and validated with high-resolution remote sensing images. Finally, three data sets

with good applicability are recommended. The data sets of CLCD, GLC30 and ESAGC are in good agreement with the yearbook data and high-resolution remote sensing images. The data set of ESAGC especially performs well in the data sets of higher spatial resolution (10 m).

## 5. Consistency Analysis and Combination of LULC Products

### 5.1. Spatial Consistency Analysis

Figure 11 shows the results of a spatial consistency analysis for the major land cover types of forest, water, cropland, impervious surface, grassland and shrubland that were obtained by overlaying the six remote sensing-based land cover data sets on each other.

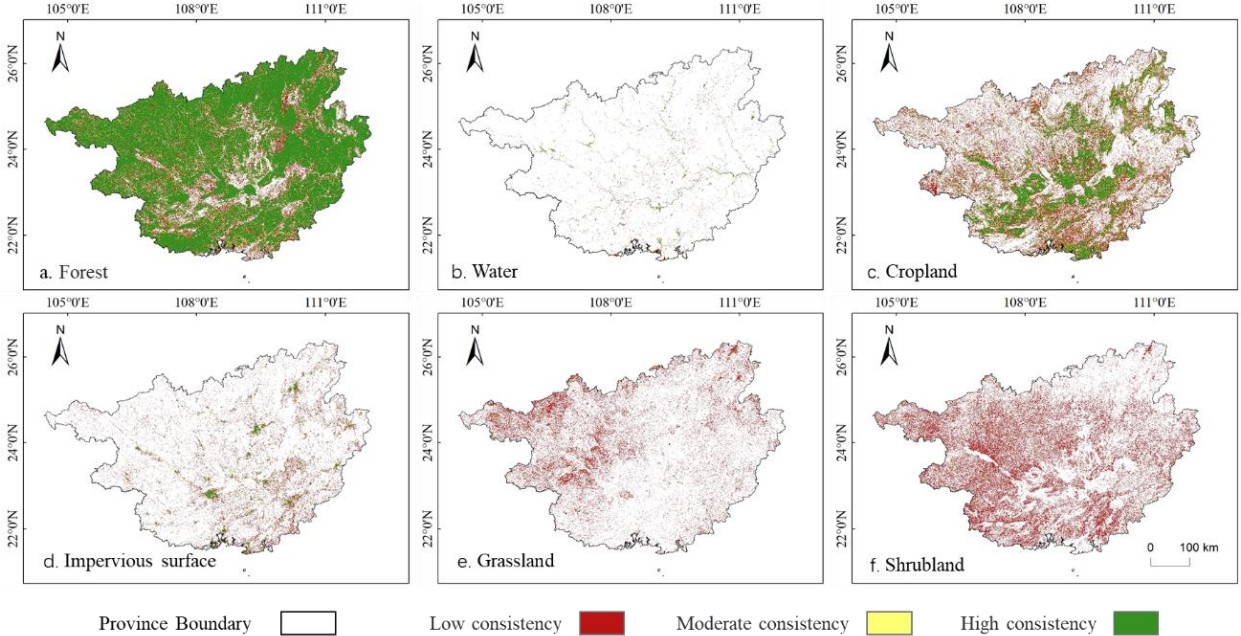

**Figure 11.** Results of spatial consistency analysis for major land cover types in Guangxi.

The six land cover data sets were found to identify the forest class best, with 84.69% of the forest area being classed as having a medium or high spatial consistency (the same areas identified as forest by more than three land cover data sets). For cropland and water, the percentage of the grid cells classed as having a medium or high spatial consistency was 51.72% and 50.66%, respectively. For the impervious surface, grassland and shrubland cover types, the percentage of cells found to have a medium or high spatial consistency was low at 27.73%, 2.32% and 0.92%, respectively. The other land cover types cover only a small area of Guangxi, and their classification is affected by many factors.

In general, in most areas of Guangxi, the six land cover data sets were found to have a high degree of spatial consistency, with areas classed as having a medium or high consistency accounting for 96.98% of the whole region (Figure 12). The areas with a high spatial consistency mainly consist of large areas of forest, cropland and water together with the urban centers. The land cover in these areas is stable and can be easily identified in remote sensing images. However, at the edge of these areas, the spatial consistency was found to be low; this is partly due to the uncertainty in pixel identification but also to the different classification standards used by the different data sets.

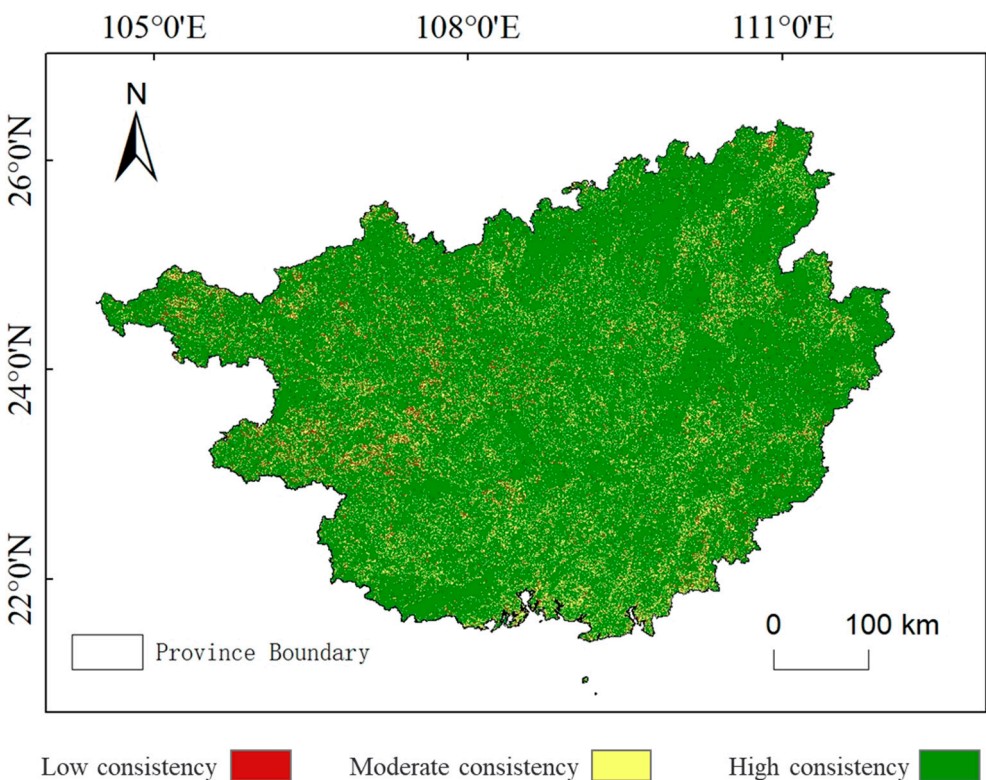

Low consistency ▮  Moderate consistency ▯  High consistency ▮

**Figure 12.** Map showing the degree of spatial consistency between the six land cover data products in the study area.

*5.2. Combination of LULC Products*

5.2.1. Combination Scheme

As the number of open-access multi-source LULC products increases, the accuracy and spatial resolution of different LULC products is continuously improving. However, currently, scientific research using LULC products at the regional scale is subject to some limitations. Because any LULC product includes errors and because inconsistencies between different LULC data sets are common, obtaining new products by fusing existing information is an effective way to solve some of these problems. Here, using the statistical yearbook data as a reference, we propose a fusion method for correcting the inconsistencies between multi-source products that can be used to obtain LULC data that better fits the reference data.

Figure 13 shows a land cover data set that was obtained by combining the six land cover data sets ESAGC, ESRI–LULC, FROM–GLC, CLCD, GLC_FCS30 and GLC30 after consideration of the evaluation results and the spatial consistency analysis described above. The following fusion rules were used: (1) for cells with a spatial consistency value greater than or equal to 3 (Three data sets belong to the same type, and the remaining three data sets belonging to another type are not included), the land cover type was determined as being the type that had the highest occurrence in the six land cover data sets. (2) In other cases, only the three data sets ESAGC, CLCD and GLC30, which had been determined to have a good performance (see Section 3), were used for the data fusion. For each cell, the land cover type was then determined as the type that had the highest occurrence in these three data sets; if the land cover type in all three data sets was different, the land cover type in ESAGC was selected. In this way, a fused land cover data set that had a high degree of spatial consistency in Guangxi was produced. The spatial distribution of the land cover types in this data set was similar to that in the six original land cover data sets.

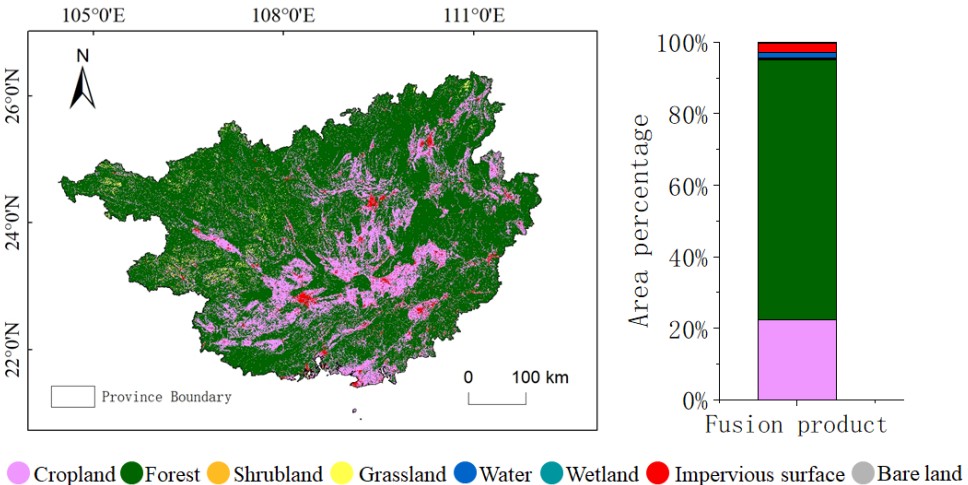

Figure 13 legend: ● Cropland ● Forest ● Shrubland ● Grassland ● Water ● Wetland ● Impervious surface ● Bare land

**Figure 13.** Map showing the spatial distribution of land cover types in Guangxi based on the six products.

### 5.2.2. Validation of the New Land Cover Product

Figure 14 shows a scatter plot of the forest area for individual cities in the fused land cover product against the statistical yearbook data for Guangxi along with box plots for the corresponding error coefficients; Figure 15 shows similar information for the cropland area for individual counties. It can clearly be seen that the performance of the fused data set is significantly better than that of the six original data sets: the fused data agree well with the statistical yearbook data, giving correlation coefficients for the city-level forest areas and county-level cropland areas of 0.998 and 0.942, respectively.

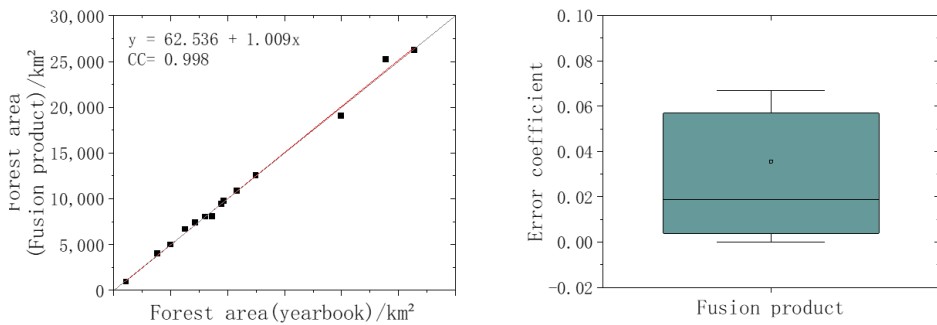

**Figure 14.** Comparison between city-level forest data and yearbook data for Guangxi with box plots of the corresponding relative errors.

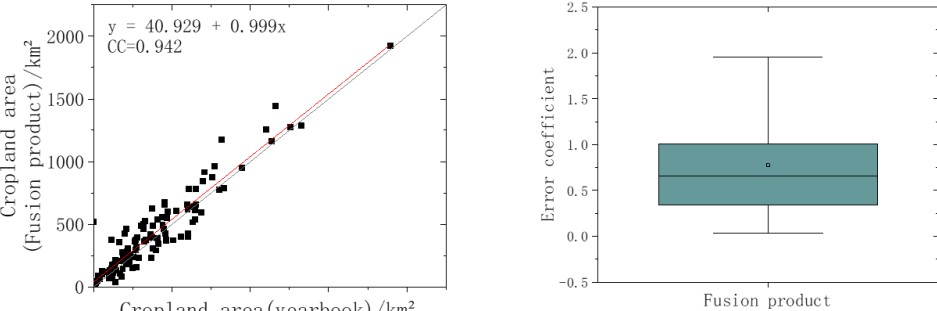

**Figure 15.** Comparison between county-level cropland data and yearbook data for Guangxi with box plots of the corresponding relative errors.

## 6. Discussion and Conclusions

### 6.1. Discussion

As mentioned above, LULC is the foundation for ecological environment research, land resource management and evaluation; thus, it could support and impact the sustainable development of the environment and society. Spatial configuration and heterogeneity of land cover could affect species distribution and biodiversity patterns [50]. The displacement of natural landscapes by man-made land cover is the most obvious aspect of urbanization [51,52]. Drawing a short-time land use–land cover map can be an important method for studying danger after disaster. LULC data can intuitively reflect the loss and transformation of forests [53]. As the core field of sustainable land use research, cultivated land resources are the basic material conditions for agricultural production. It is related to food security, human survival and development [53–57]. Ensuring land management systems that maintain land cover is critical to sustaining human livelihoods in Africa [58]. LULC variation is one of the important factors affecting runoff in high-altitude catchment areas, which is of great significance for sustainable management and ecological development of water resources [59]. Therefore, the accuracy of land cover data will affect the indicators corresponding to the sustainable development goals [60,61].

The data accuracy assessment in this study was based on regional data concerning cropland, forest, grassland and other land cover types, and most existing reliability assessments of global LUCC data sets were made in this way. At present, the quantitative evaluation method based on statistical yearbook data could only be applied to limited areas with good data availability [33,35]. This method would have limited applicability in areas where accurate historical data is lacking. Consistency evaluation is a reliable method for evaluating multiple data sets without the use of objective reference standards. This method assumes that the data and methods used to produce the different data sets are reasonable; each original data set has certain limitations but also approximates the "true value" to some degree. The higher the consistency, the more likely the data are to be reliable [24,34]. In data evaluation, how to objectively and reasonably determine the discrimination criteria and how to extend regional studies to larger spatiotemporal scales needs further in-depth research [62,63].

The consistency evaluation results showed that the consistency of shrub and bare land was poor, which might be caused by the difference in the definition of land cover type. It may be necessary to limit the definition of certain land cover type based on a series of measurable features. This can make the classification more accurate, but too fine classification may also lead to the phenomenon of feature redundancy. How to reasonably establish classification standards is also the direction to be discussed. Moreover, the satellite data, processing methods, classification systems and classification methods used to produce the different land cover products are also different, which could lead to the poor consistency of certain land cover types [18,23]. Land cover classification systems are often aimed at specific research purposes and research scales, and there is no uniform standard. As a result, the differences between the different classification systems could limit comparative evaluations between different land cover products.

### 6.2. Conclusions

In this study, the performance and consistency of six large-scale remote sensing land cover products were evaluated and analyzed at the provincial scale in Guangxi. The methods used were based mainly on the use of statistical data for specific land cover types obtained from regional yearbooks; a new fused data set was also produced following a spatial consistency analysis. The original data sets that were used included ESA GlobCover (ESAGC), Esri Land Use/Land Cover (ESRI–LULC), Finer Resolution Observation and Monitoring of Global Land Cover (FROM–GLC), the China Land Cover Dataset (CLCD), the Global Land Cover product with Fine Classification System at 30 m (GLC_FCS30) and GlobeLand30 (GLC30).

The main conclusions of this research are as follows.

(1)     Based on consideration of the spatial distribution of different land cover types, there is good agreement between the six different LULC data sets in Guangxi. FROM–GLC and GLC30 data are the most highly correlated with each other. For any two products, the correlation coefficient between the areas covered by a given land cover type is greater than 0.828, with a maximum value of 0.999. All six data sets show that forest and cropland are the two main land cover types in Guangxi.

(2)     The data in ESRI–LULC, CLCD and GLC30 compare well with the forest data in the statistical yearbooks, whereas ESAGC, CLCD and GLC30 are in good agreement with the cropland data in these yearbooks. By comparing the details with Google Earth images, ESAGC, CLCD and GLC30 are of high precision and accuracy in Guangxi, China. Especially, the data set of ESAGC performs well in the data sets of higher spatial resolution (10 m).

(3)     The six land use–land cover data sets have a high degree of spatial consistency in most regions of Guangxi, with the areas of medium and high consistency accounting for 96.98% of the total area. Overall, the six data sets have the highest degree of consistency for areas of forest, followed by water bodies and cropland.

(4)     The fused data set that was produced following the spatial consistency analysis agrees better with the cropland and forest data in the statistical yearbooks than any of the six original data sets.

Research into the accuracy of land cover–land use data at the regional scale is a necessary prerequisite for the application of these data to support regional sustainable development. The results of this study provide a basis for the application of global or national land cover data at regional scales. The fused land cover data product that was produced can be used to provide support for land management and environmental research in Guangxi. Accurate land resource utilization data can effectively support the management and strategic use of land resources by relevant government agencies by addressing the problems that result from a lack of traditional data or inaccurate data, thereby providing support for applied research into regional sustainable development.

**Author Contributions:** Conceptualization, Y.Q. and X.H.; methodology, G.J. and X.H.; investigation, Y.Q., G.J. and X.H.; writing—original draft preparation, X.H.; writing—review and editing, X.H., Y.Q., G.J. and M.M.; visualization, X.H. and Z.J.; supervision, Y.Q. and J.M.; funding acquisition, Y.Q. and J.M. All authors have read and agreed to the published version of the manuscript.

**Funding:** This study was supported by the Innovation Driven Development Special Project of Guangxi (GuikeAA20302022), the Strategic Priority Research Program of the Chinese Academy of Sciences (XDA19090130) and the Guangxi Key Research and Development Program (AB21220057). Authors acknowledges the support by the Chinese Academy of Sciences President's International Fellowship Initiative (Grant No. 2021VTA0007 and No. 2020VTA0001) and the Ministry of Science and Technology High Level Foreign Expert Program (Grant No. GL20200161002).

**Data Availability Statement:** ESA GlobCover is from European Space Agency (https://esa-worldcover.org/en, accessed on 12 March 2022); ESRI Land Use/Land Coveris is developed by ESRI (https://www.arcgis.com/apps/instant/media/index.html?appid=fc92d38533d440078f17678ebc20e8e2, accessed on 12 March 2022); FROM-GLC10 is from Tsinghua University. (http://data.ess.tsinghua.edu.cn/fromglc2017v1.html, accessed on 12 March 2022); GLC_FCS30 is from Aerospace Information Research Institute, Chinese Academy of Sciences (http://data.casearth.cn/sdo/detail/5fbc7904819aec1ea2dd7061, accessed on 12 March 2022); GlobeLand30 is from National Geomatics Center of China (http://mulu.tianditu.gov.cn/mapDataAction.do?method=globalLandCover, accessed on 12 March 2022); Annual city-level forest area statistics and county-level crop-area statis-tics from Guangxi Zhuang Autonomous Region Bureau of Statistics (http://tjj.gxzf.gov.cn/, accessed on 5 January 2022).

**Conflicts of Interest:** The authors declare no conflict of interest.

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
