# Peer review of "Evaluation of Global Land Use–Land Cover Data Products in Guangxi, China"

_remotesensing, doi:10.3390/rs15051291_

Round 1
Reviewer 1 Report
This paper aims to assess the quality of six popular global land cover products in the local area. They collected six global land cover maps and analyzed their performance according to forest and cropland distribution from yearbook data in different regions. The results presented in this paper sound reasonable. But I still have some concerns before this manuscript is recommended for publication.
1). As we know, forest, cropland, and build-up are the main land cover types in Guangxi and their distribution is also important for SDGs, so if only these three types and their areas are considered, which product is recommended? I suggest highlighting this point.
2). Some references should be double-checked carefully. For example, this paper was not cited correctly.
Chen B, Xu B, Zhu Z, et al. (2019). Stable classification with limited sample: Transferring a 30-m resolution sample set collected in 2015 to mapping 10-m resolution global land cover in 2017. Science Bulletin, 64: 370-373.
It is not in the right format, because the authors and their orders are wrong.
Author Response
Dear Editor and Reviewer,
Thank you very much for your valuable comments and suggestions on our manuscript. we have modified and improved our manuscript according to your kind advices and detailed suggestions. The performance and applicability of global data in local scale matters, and we try to provide a case of land cover data. We sincerely hope this manuscript will be acceptable to be published on Remote Sensing.
Have a nice day and look forward to your reply soon.
Best regard
Xuan Hao

Reviewer 2 Report
Manuscript details:
Journal: Remote Sensing
Manuscript ID: remotesensing-2106743
Type of manuscript: Article
Title: Evaluation of global land use/land cover data products in Guangxi, China.
Authors:
Xuan Hao, Yubao Qiu *, Guoqiang Jia, Massimo Menenti, Jiangming Ma, and Zhengxin Jiang
I read with great interest the manuscript submitted by Xuan Hao et al. for consideration of Remote sensing. This manuscript presents a great idea regarding to improvement of land use/land cover (LULC) by the performance of six large-scale remote sensing land cover products were evaluated and analyzed at the provincial scale in Guangxi. For me, the article is acceptable; the objectives are clear, and the results are well presented. The manuscript needs to be reviewed by an English language specialist, there are some long sentences and some repetitive sentence links. Moreover, some punctuation marks need review. References in the text need review, the research paper relies on the data of six data sources compared to the statistical yearbook, which needs to mention some data on the accuracy of the yearbook data compared to the 6 data mentioned in the manuscript. Hereunder my comments. I hope to contribute to the improvement of the manuscript along with the comments of other reviewers.
Abstract
The abstract is very big (350 words) and needs to be 200-250 words. Also, the abstract needs to be written in a better way, its language is not good, and it needs direct phrases.
Keywords: Please replace (LULC) with the full-term Land use/land cover. There is no need to (fusion).
1. Introduction
- Page 2, first paragraph, Repeating the word (and) makes the sentences weak. Try to rephrase the sentences to sound more correct and more accurate by reviewing from a specialist or a native speaker.
- Page 2, first paragraph, last two lines with 6 references, please arrange them alphabetically or by date.
- Page 2, last line (UMD) please write the full words and then the abbreviation.
- Page 3, second paragraph, (Yang et al. made a qualitative) Kindly add date.
- Page 3, second paragraph, there are three references that need to be dated.
- Page 3, First paragraph, last line (N Breznau et al.,2021) it is enough to write the family name.
- Page 4, second paragraph, second line, the authors may consider using decimal degrees.
- Page 4, It would be good to add a map of the study area showing the topography and the geomorphological units or a land use map, to show some physical and human geography of the study area.
2. Data and methods
- Page 5, paragraph 2, It is better to put a reference at the end of the paragraph, and also include a link to the web page of ESA GlobCover.
- Page 5, paragraph 2, It is better to put a reference at the end of the paragraph, and also include a link to the web page of ESRI Land Use/Land Cover.
- Page 5, the same note as in the rest of the paragraphs.
- A very important question: How did the authors obtain the accuracy of the data? What if there were previous studies that indicated accuracy? Please clarify this point because the whole manuscript is based on it.
- Page 6, Table 1, third column, last row. The authors mentioned on the previous page that it is (Overall Accuracy) 85% and in table 1 86%.
- Page 7, Line 1, something is missing in this sentence.
- Page 7, subtitle (2.2 Consolidation of LULC Classification) References order date or alphabetical.
- Page 9, subtitle (2.3.1 Correlation Analysis), Equation 1, are there other studies around the world that applied this equation to be cited by the authors or is it from the authors, please explain.
- Page 9, subtitle (2.3.2 Accuracy Assessment), Equation 2 , to make accuracy assessment there are many equations, please mention some similar studies that applied such an equation to test the accuracy of its data.
- Page 9 line 13, which version of statistical yearbook? Please mention to it and the website if it's possible.
3. LULC Data Product Reclassification
- Page 10, subtitle 3.1 Spatial Distribution of Land Cover Types in Reclassified LULC Datasets, line 33, It is preferable to mention some details about the reclassification mechanism.
- Page 10, Figure 1. It is very important to mention the date of the data.
4. Evaluation of the Accuracy of LULC Products
- Page 12 fig 3, (C. LULC) I think it's better to be complete the form (ESRI-LULC). Also, Fig 3g doesn’t mention in the text.
- Page 14 fig 6 (C. LULC) it's better to be complete the form (ESRI-LULC). Also, Fig 6g doesn’t mention in the text.
- Page 15, line 122, Please mention the accuracy of Google Earth data.
- Page 16, fig 9 (a) It is enough to put the numbers of places without the Latin numbering.
5. Consistency Analysis and Combination of LULC Products
- Page 17, Figure 10. This shape needs to be simplified or more clarified, maybe it needs a colour change.
- Page 17, Figure 10. It's best to merge the categories into just 3 to make it easier to differentiate (low- moderate- high).
6. Discussion and Conclusion
- The discussion section is missing some clarification and connection with the aims of the manuscript. Specifically regarding the relationship of sustainable development with accurate data and improving the production of LULC data. With mentioning previous studies and research around the world to support the idea that this research aims for. By comparing the idea with similar ones around the world.
References
- Some of reference needs revision for example: Verburg P H, Erb K H, Mertz O, et al., (2013) and also, Fan Y T,Jin X B,Xiang X M et al.,(2019). Please write the full list of authors of these papers.
- The reference list needs to be one style for example these references (Breznau, N., Rinke, E. M., Wuttke, A., Nguyen, H. H. V., Adem, M., Adriaans, J.,) Some authors have a family name and a letter, and some have a family name and two letters. Please standardize the method.
Author Response
Dear Editor and Reviewer,
Thank you very much for your valuable comments and suggestions on our manuscript. we have modified and improved our manuscript according to your kind advices and detailed suggestions. The performance and applicability of global data in local scale matters, and we try to provide a case of land cover data. We sincerely hope this manuscript will be acceptable to be published on Remote Sensing.
Please see the attachment.
Have a nice day and look forward to your reply soon.
Best regard
Xuan Hao

Reviewer 3 Report
Comments to the Author
Review of Article “Evaluation of global land use/land cover data products in Guangxi, China” by Xuan HAO, Yubao QIU, Guoqiang JIA, Massimo Menenti, Jiangming MA and Zhengxin JIANG
This article deals with the evaluation of six state-of-the-art LULC data products in Guangxi, China.
In my opinion, this work is interesting but the authors should improve and, discuss a bit more some points. I believe that the authors should provide clearer some points in the section of “Data and Methods”. In my opinion, the authors should provide some clarification in some points over the text.
I believe that this article has the potential to be published in “Remote Sensing” Journal after a major revision. The authors could take under consideration the suggestions and comments below.
1. I suggest the authors make clear in this section the novelty of this work and also to give clearer why this work is important.
2. In page 3 line 57 the authors said that “The correlation between FROM-GLC and GLC30 was the best – giving a correlation coefficient of 0.999 . . . “. Is this a spatial correlation? Could you please clarify this point?
3. In page 3 line 60 the authors said that “On the whole, the correlation between 59 GLC_FCS30 and the other five datasets was weak”. What do you mean “weak”, a value less than 0.4 for example?
4. In Figure 3a I suggest the authors include the names in axes x.
5. In page 4 the authors said that “The forest areas in the LULC, FROM-GLC, CLCD and GLC30 datasets are close to the values given in the yearbooks”. Can you give an explanation about this?
6. In Figure 6 please see my previous comment 3.
7. In page 7 lines 120-121 the authors said that “We then selected 18 sample areas for verification in Guangxi (Figure 9a); these included areas of cropland, forest and complex land cover types”. Why do you select these regions? Could you please discuss this point?
8. In my opinion, the authors should provide some more elements in “2.3.3 Spatial Consistency Analysis” section. I suggest to provide more elements regarding this method (3 or 4 sentences) in order to make it clearer to the reader.
9. The title in paragraph 4.3 is “Comparison and Verification Based on High-resolution Remote Sensing Images”. Maybe I am missing something but it is not clear to me the comparison method that the author used in this section. Could you please clarify this point?
10. In “discussion” section, I am waiting the authors’ discussion regarding their results and also the presentation of some points from literature. I suggest the authors give some points from their main findings and a discussion based on literature.
11. In page 20 line 232-234 the authors said that “Based on a 232 comparison between the six datasets and high-resolution remote sensing images obtained from Google Earth, the performance of ESAGC, CLCD and GLC30 is generally good in Guangxi, China.” Could you please make clear what “the performance of ESAGC, CLCD and GLC30 is generally good in Guangxi” means?
Minor Comments:
1. In Introduction section, in sentence “They recommend that the use of global LULC products should involve critical evaluation of their suitability with respect to the application purpose(Venter al., 2022).” There is a missing gap in “purpose(Venter al., 2022)”.
2. In Introduction section, in the paragraph “For example, Yang et al. made a qualitative and quantitative analysis and comparison of the five existing kinds of land cover data for China and the surrounding areas based on a spatial consistency analysis and by sampling the high resolution images in Google Earth The results revealed large areas where there was disagreement between the five land cover datasets and the overall consistency was low (Yang et al., 2014)” a “.” is missing in “…Google Earth The results…”
3. In page 7 line 124-125 there is an extra gap.
4. In page 8 line 126 a full stop is missing.
Author Response

(The authors gave the same response as above.)

Reviewer 4 Report
Review report for
Evaluation of global land use/land cover data products in Guangxi, China
1. Please briefly summarise the research gap in the Abstract.
2. Please illustrate the full description of the ESA in the Abstract.
3. Please give the full description of the following abbreviations in the Introduction.
a. UMD
b. IGBP
c. MODIS
d. ESA
e. ESRI
4. In the Introduction, the literature review in the second-last paragraph concentrates on analysing research results from 2010 to 2019.
The reader needs to know the current state of research for the last five years. Therefore, please include some reviews of quantitative studies of land cover information at the regional level in the previous five years.
5. In the Introduction, please illustrate a precise research aim and question.
6. In the Introduction, only the reasons for choosing Guangxi as a case study must be briefly given. Currently, the last paragraph of the Introduction is too wordy. The case study needs to be presented as a separate section. Please consider placing this paragraph in the second section, Data and Methods.
7. In section 2.1.1, please indicate why the six commonly used datasets were chosen. Why these six datasets?
8. Section 2.3 deals with two formulas. What are the sources of these two formulas?
9. Please revise the term "stacking number" in sub-section 2.3.3.
10. Please ensure all references are in the correct order and format.
Author Response

(The authors gave the same response as above.)

Round 2
Reviewer 3 Report
The updated version of this work has been improved. The authors have answered the reviewer's comments and suggestions.
I suggest the publication of the article in the "Remote Sensing" Journal.